# Efficient and Accurate Estimation of Lipschitz Constants for Deep Neural Networks

**Mahyar Fazlyab**
ESE Department
University of Pennsylvania
Philadephia , PA 19104
mahyarfa@seas.upenn.edu

**Alexander Robey**
ESE Department
University of Pennsylvania
Philadephia , PA 19104
arobey1@seas.upenn.edu

**Hamed Hassani**
ESE Department
University of Pennsylvania
Philadephia , PA 19104
hassani@seas.upenn.edu

**Manfred Morari**
ESE Department
University of Pennsylvania
Philadephia , PA 19104
morari@seas.upenn.edu

**George J. Pappas**
ESE Department
University of Pennsylvania
Philadephia , PA 19104
pappasg@seas.upenn.edu

## Abstract

Tight estimation of the Lipschitz constant for deep neural networks (DNNs) is useful in many applications ranging from robustness certification of classifiers to stability analysis of closed-loop systems with reinforcement learning controllers. Existing methods in the literature for estimating the Lipschitz constant suffer from either lack of accuracy or poor scalability. In this paper, we present a convex optimization framework to compute guaranteed upper bounds on the Lipschitz constant of DNNs both accurately and efficiently. Our main idea is to interpret activation functions as gradients of convex potential functions. Hence, they satisfy certain properties that can be described by quadratic constraints. This particular description allows us to pose the Lipschitz constant estimation problem as a semidefinite program (SDP). The resulting SDP can be adapted to increase either the estimation accuracy (by capturing the interaction between activation functions of different layers) or scalability (by decomposition and parallel implementation). We illustrate the utility of our approach with a variety of experiments on randomly generated networks and on classifiers trained on the MNIST and Iris datasets. In particular, we experimentally demonstrate that our Lipschitz bounds are the most accurate compared to those in the literature. We also study the impact of adversarial training methods on the Lipschitz bounds of the resulting classifiers and show that our bounds can be used to efficiently provide robustness guarantees.

## 1 Introduction

A function $f: \mathbb{R}^n \to \mathbb{R}^m$ is globally Lipschitz continuous on $\mathcal{X} \subseteq \mathbb{R}^n$ if there exists a nonnegative constant $L \geq 0$ such that

$$\|f(x) - f(y)\| \leq L\|x - y\| \text{ for all } x, y \in \mathcal{X}. \tag{1}$$

The smallest such $L$ is called *the* Lipschitz constant of $f$. The Lipschitz constant is the maximum ratio between variations in the output space and variations in the input space of $f$ and thus is a measure of sensitivity of the function with respect to input perturbations.

When a function $f$ is characterized by a deep neural network (DNN), tight bounds on its Lipschitz constant can be extremely useful in a variety of applications. In classification tasks, for instance, $L$ can be used as a certificate of robustness of a neural network classifier to adversarial attacks if

it is estimated tightly [34]. In deep reinforcement learning, tight bounds on the Lipschitz constant of a DNN-based controller can be directly used to analyze the stability of the closed-loop system. Lipschitz regularity can also play a key role in derivation of generalization bounds [6]. In these applications and many others, it is essential to have tight bounds on the Lipschitz constant of DNNs. However, as DNNs have highly complex and non-linear structures, estimating the Lipschitz constant both accurately and efficiently has remained a significant challenge.

**Our contributions.** In this paper we propose a novel convex programming framework to derive tight bounds on the global Lipschitz constant of deep feed-forward neural networks. Our framework yields significantly more *accurate* bounds compared to the state-of-the-art and lends itself to a distributed implementation, leading to *efficient* computation of the bounds for large-scale networks.

**Our approach.** We use the fact that all common nonlinear activation functions used in neural networks are gradients of convex functions; hence, as operators, they satisfy certain properties that can be abstracted as quadratic constraints on their input-output values. This particular abstraction allows us to pose the Lipschitz estimation problem as a semidefinite program (SDP), which we call LipSDP. A striking feature of LipSDP is its flexibility to span the trade-off between estimation accuracy and computational efficiency by adding or removing extra decision variables. In particular, for a neural network with $\ell$ layers and a total of $n$ hidden neurons, the number of decision variables can vary from $\ell$ (least accurate but most scalable) to $O(n^2)$ (most accurate but least scalable). As such, we derive several distinct yet related formulations of LipSDP that span this trade-off. To scale each variant of LipSDP to larger networks, we also propose a distributed implementation.

**Our results.** We illustrate our approach in a variety of experiments on both randomly generated networks as well as networks trained on the MNIST [23] and Iris [11] datasets. First, we show empirically that our Lipschitz bounds are the most accurate compared to all other existing methods of which we are aware. In particular, our experiments on neural networks trained for MNIST show that our bounds outperform all comparable methods; see Figure 2a for details. Furthermore, we investigate the effect of two robust training procedures [24, 40] on the Lipschitz constant for networks trained on the MNIST dataset. Our results suggest that robust training procedures significantly decrease the Lipschitz constant of the resulting classifiers. Moreover, we use the Lipschitz bound for two robust training procedures to derive non-vacuous lower bounds on the minimum adversarial perturbation necessary to change the classification of any instance from the test set. For details, see Figure 3.

**Related work.** The problem of estimating the Lipschitz constant for neural networks has been studied in several works. In [34], the authors estimate the global Lipschitz constant of DNNs by the product of Lipschitz constants of individual layers. This approach is scalable and general but yields trivial bounds. In [10], the authors derive bounds on Lipschitz constants by treating the activation functions as non-expansive averaged operators. The resulting algorithm scales well with the number of hidden units per layer, but exponentially with the number of layers. In [37], the authors decompose the weight matrices of a neural network via singular value decomposition and approximately solve a convex maximization problem over the unit cube. Notably, estimating the Lipschitz constant using the method in [37] is intractable even for small networks; indeed, the authors of [37] use a greedy algorithm to compute a bound, which may underapproximate the Lipschitz constant. In [2], the maximum spectral norm of the network Jacobian (taken over the data distribution) is used as an estimate of the true Lipschitz constant. Again, this approach is not guaranteed to be an upper bound on the Lipschitz constant. Bounding Lipschitz constants for the specific case of convolutional neural networks (CNNs) has also been addressed in [5, 44, 6].

Using Lipschitz bounds in the context of adversarial robustness and safety verification has also been addressed in several works [39, 31, 38]. In particular, in [39], the authors convert the robustness analysis problem into a local Lipschitz constant estimation problem, where they estimate this local constant by a set of independently and identically sampled local gradients. This algorithm is scalable but is not guaranteed to provide upper bounds. In a similar work, the authors of [38] exploit the piece-wise linear structure of ReLU functions to estimate the local Lipschitz constant of neural networks. In [13], the authors use quadratic constraints and semidefinite programming to analyze local (point-wise) robustness of neural networks. In contrast, our Lipschitz bounds can be used as a global certificate of robustness and are agnostic to the choice of the test data.

## 1.1 Motivating applications

We now enumerate two applications that highlight the importance of estimating the Lipschitz constant of DNNs accurately and efficiently.

**Robustness certification of classifiers.** In response to fragility of DNNs to adversarial attacks, there has been considerable effort in recent years to improve the robustness of neural networks against adversarial attacks and input perturbations [16, 26, 43, 22, 24, 40]. In order to certify and/or improve the robustness of neural networks, one must be able to bound the possible outputs of the neural network over a region of input space. This can be done either locally around a specific input [7, 35, 15, 33, 12, 29, 30, 13, 40, 21, 41, 42], or globally by bounding the sensitivity of the function to input perturbations, i.e., the Lipschitz constant [19, 34, 28, 39]. Indeed, tight upper bounds on the Lipschitz constant can be used to derive non-vacuous lower bounds on the magnitudes of perturbations necessary to change the decision of neural networks. Finally, an efficient computation of these bounds can be useful in either assessing robustness after training [29, 30, 13] or promoting robustness during training [40, 36, 17]. In the experiments section, we explore this application in depth.

**Stability analysis of closed-loop systems with learning controllers.** A central problem in learning-based control is to provide stability or safety guarantees for a feedback control loop when a learning-enabled component, such as a deep neural network, is introduced in the loop [4, 8, 20]. The Lipschitz constant of a neural network controller bounds its gain. Therefore a tight estimate can be useful for certifying the stability of the closed-loop system.

**Notation.** We denote the set of real $n$-dimensional vectors by $\mathbb{R}^n$, the set of $m \times n$-dimensional matrices by $\mathbb{R}^{m \times n}$, and the $n$-dimensional identity matrix by $I_n$. We denote by $\mathbb{S}^n$, $\mathbb{S}^n_+$, and $\mathbb{S}^n_{++}$ the sets of $n$-by-$n$ symmetric, positive semidefinite, and positive definite matrices, respectively. The $p$-norm ($p \geq 1$) is denoted by $\| \cdot \|_p \colon \mathbb{R}^n \to \mathbb{R}_+$. The $\ell_2$-norm of a matrix $W \in \mathbb{R}^{m \times n}$ is the largest singular value of $W$. We denote the $i$-th unit vector in $\mathbb{R}^n$ by $e_i$. We write $\mathrm{diag}(a_1, ..., a_n)$ for a diagonal matrix whose diagonal entries starting in the upper left corner are $a_1, \cdots, a_n$.

# 2 LipSDP: Lipschitz certificates via semidefinite programming

## 2.1 Problem statement

Consider an $\ell$-layer feed-forward neural network $f(x) \colon \mathbb{R}^{n_0} \to \mathbb{R}^{n_{\ell+1}}$ described by the following recursive equations:

$$x^0 = x, \quad x^{k+1} = \phi(W^k x^k + b^k) \text{ for } k = 0, \cdots, \ell - 1, \quad f(x) = W^\ell x^\ell + b^\ell. \tag{2}$$

Here $x \in \mathbb{R}^{n_0}$ is an input to the network and $W^k \in \mathbb{R}^{n_{k+1} \times n_k}$ and $b^k \in \mathbb{R}^{n_{k+1}}$ are the weight matrix and bias vector for the $k$-th layer. The function $\phi$ is the concatenation of activation functions at each layer, i.e., it is of the form $\phi(x) = [\varphi(x_1) \cdots \varphi(x_n)]^\top$. In this paper, our goal is to find tight bounds on the Lipschitz constant of the map $x \mapsto f(x)$ in $\ell_2$-norm. More precisely, we wish to find the smallest constant $L_2 \geq 0$ such that $\|f(x) - f(y)\|_2 \leq L_2 \|x - y\|_2$ for all $x, y \in \mathbb{R}^{n_0}$.

The main source of difficulty in solving this problem is the presence of the nonlinear activation functions. To combat this difficulty, our main idea is to abstract these activation functions by a set of constraints that they impose on their input and output values. Then any property (including Lipschitz continuity) that is satisfied by our abstraction will also be satisfied by the original network.

## 2.2 Description of activation functions by quadratic constraints

In this section, we introduce several definitions and lemmas that characterize our abstraction of nonlinear activation functions. These results are crucial to the formulation of an SDP that can bound the Lipschitz constants of networks in Section 2.3.

**Definition 1 (Slope-restricted non-linearity)** *A function $\varphi \colon \mathbb{R} \to \mathbb{R}$ is slope-restricted on $[\alpha, \beta]$ where $0 \leq \alpha < \beta < \infty$ if*

$$\alpha \leq \frac{\varphi(y) - \varphi(x)}{y - x} \leq \beta \quad \forall x, y \in \mathbb{R}. \tag{3}$$

The inequality in (3) simply states that the slope of the chord connecting any two points on the curve of the function $x \mapsto \varphi(x)$ is at least $\alpha$ and at most $\beta$ (see Figure 1). By multiplying all sides of (3) by $(y-x)^2$, we can write the slope restriction condition as $\alpha(y-x)^2 \leq (\varphi(y)-\varphi(x))(y-x) \leq \beta(y-x)^2$. By the left inequality, the operator $\varphi(x)$ is strongly monotone with parameter $\alpha$ [32], or equivalently the anti-derivative function $\int \varphi(x)dx$ is strongly convex with parameter $\alpha$. By the right-hand side inequality, $\varphi(x)$ is one-sided Lipschitz with parameter $\beta$. Altogether, the preceding inequalities state that the anti-derivative function $\int \varphi(x)dx$ is $\alpha$-strongly convex and $\beta$-smooth.

Note that, except for special cases [2], all common activation functions used in deep learning satisfy the slope restriction condition in (3) for some $0 \leq \alpha < \beta < \infty$. For instance, the ReLU, tanh, and sigmoid activation functions are all slope restricted with $\alpha = 0$ and $\beta = 1$. More details can be found in [13].

**Definition 2 (Incremental Quadratic Constraint [1])** A function $\phi \colon \mathbb{R}^n \to \mathbb{R}^n$ satisfies the incremental quadratic constraint defined by $\mathcal{Q} \subset \mathbb{S}^{2n}$ if for any $Q \in \mathcal{Q}$ and $x, y \in \mathbb{R}^n$,

$$\begin{bmatrix} x - y \\ \phi(x) - \phi(y) \end{bmatrix}^\top Q \begin{bmatrix} x - y \\ \phi(x) - \phi(y) \end{bmatrix} \geq 0. \tag{4}$$

In the above definition, $\mathcal{Q}$ is the set of all *multiplier matrices* that characterize $\phi$, and is a convex cone by definition. As an example, the softmax operator $\phi(x) = (\sum_{i=1}^n \exp(x_i))^{-1}[\exp(x_1) \cdots \exp(x_n)]^\top$ is the gradient of the convex function $\psi(x) = \log(\sum_{i=1}^n \exp(x_i))$. This function is smooth and strongly convex with paramters $\alpha = 0$ and $\beta = 1$ [9]. For this class of functions, it is known that the gradient function $\phi(x) = \nabla \psi(x)$ satisfies the quadratic inequality [25]

$$\begin{bmatrix} x - y \\ \phi(x) - \phi(y) \end{bmatrix}^\top \begin{bmatrix} -2\alpha\beta I_n & (\alpha + \beta)I_n \\ (\alpha + \beta)I_n & -2I_n \end{bmatrix} \begin{bmatrix} x - y \\ \phi(x) - \phi(y) \end{bmatrix} \geq 0. \tag{5}$$

Therefore, the softmax operator satisfies the incremental quadratic constraint defined by $\mathcal{Q} = \{\lambda M \mid \lambda \geq 0\}$, where $M$ the middle matrix in the above inequality.

To see the connection between incremental quadratic constraints and slope-restricted nonlinearities, note that (3) can be equivalently written as the single inequality

$$(\frac{\varphi(y) - \varphi(x)}{y - x} - \alpha)(\frac{\varphi(y) - \varphi(x)}{y - x} - \beta) \leq 0. \tag{6}$$

Multiplying through by $(y - x)^2$ and rearranging terms, we can write (6) as

$$\begin{bmatrix} x - y \\ \varphi(x) - \varphi(y) \end{bmatrix}^\top \begin{bmatrix} -2\alpha\beta & \alpha + \beta \\ \alpha + \beta & -2 \end{bmatrix} \begin{bmatrix} x - y \\ \varphi(x) - \varphi(y) \end{bmatrix} \geq 0, \tag{7}$$

which, in view of Definition 2, is an incremental quadratic constraint for $\varphi$. From this perspective, incremental quadratic constraints generalize the notion of slope-restricted nonlinearities to multi-variable vector-valued nonlinearities.

**Repeated nonlinearities.** Now consider the vector-valued function $\phi(x) = [\varphi(x_1) \cdots \varphi(x_n)]^\top$ obtained by applying a slope-restricted function $\varphi$ component-wise to a vector $x \in \mathbb{R}^n$. By exploiting the fact that the same function $\varphi$ is applied to each component, we can characterize $\phi(x)$ by $O(n^2)$ incremental quadratic constraints. In the following lemma, we provide such a characterization.

**Lemma 1** *Suppose $\varphi \colon \mathbb{R} \to \mathbb{R}$ is slope-restricted on $[\alpha, \beta]$. Define the set*

$$\mathcal{T}_n = \{T \in \mathbb{S}^n \mid T = \sum_{i=1}^n \lambda_{ii} e_i e_i^\top + \sum_{1 \leq i < j \leq n} \lambda_{ij}(e_i - e_j)(e_i - e_j)^\top, \lambda_{ij} \geq 0\}. \tag{8}$$

*Then for any $T \in \mathcal{T}_n$ the vector-valued function $\phi(x) = [\varphi(x_1) \cdots \varphi(x_n)]^\top \colon \mathbb{R}^n \to \mathbb{R}^n$ satisfies*

$$\begin{bmatrix} x - y \\ \phi(x) - \phi(y) \end{bmatrix}^\top \begin{bmatrix} -2\alpha\beta T & (\alpha + \beta)T \\ (\alpha + \beta)T & -2T \end{bmatrix} \begin{bmatrix} x - y \\ \phi(x) - \phi(y) \end{bmatrix} \geq 0 \ \textit{for all } x, y \in \mathbb{R}^n. \tag{9}$$

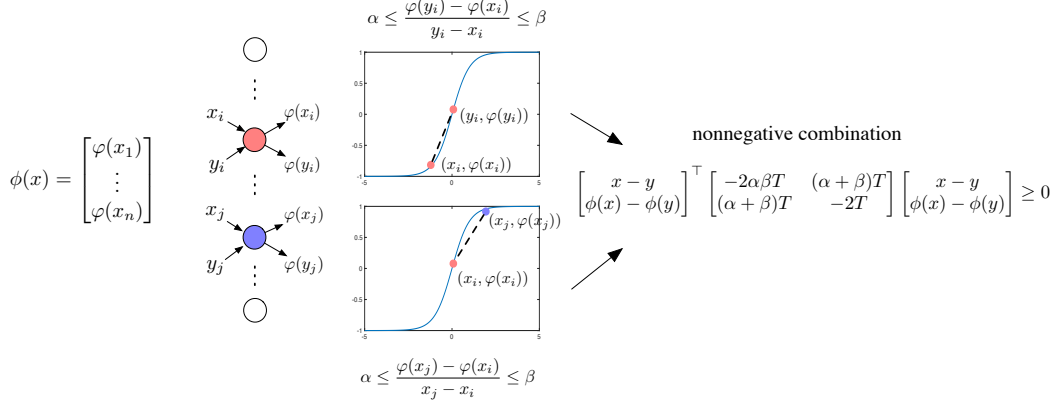

**Figure 1:** An illustrative description of encoding activation functions by quadratic constraints.

Concretely, this lemma captures the coupling between neurons in a neural network by taking advantage of two particular structures: (a) the same activation function is applied to each hidden neuron and (b) all activation functions are slope-restricted on the same interval $[\alpha, \beta]$. In this way, we can write the slope restriction condition in (1) for any pair of activation functions in a given neural network. A conic combination of these constraints would yield (9), where $\lambda_{ij}$ are the coefficients of this combination. See Figure 1 for an illustrative description.

We will see in the next section that the matrix $T$ that parameterizes the multiplier matrix in (9) appears as a decision variable in an SDP, in which the objective is to find an admissible $T$ that yields the tightest bound on the Lipschitz constant.

### 2.3 LipSDP for single-layer neural network

To develop an optimization problem to estimate the Lipschitz constant of a fully-connected feed-forward neural network, the key insight is that the Lipschitz condition in (1) is in fact equivalent to an incremental quadratic constraint for the map $x \mapsto f(x)$ characterized by the neural network. By coupling this to the incremental quadratic constraints satisfied by the cascade combination of the activation functions [14], we can develop an SDP to minimize an upper bound on the Lipschitz constant of $f$. This result is formally stated in the following theorem.

**Theorem 1 (Lipshitz certificates for single-layer neural networks)** *Consider a single-layer neural network described by* $f(x) = W^1\phi(W^0 x + b^0) + b^1$. *Suppose* $\phi(x) \colon \mathbb{R}^n \to \mathbb{R}^n = [\varphi(x_1) \cdots \varphi(x_n)]$, *where* $\varphi$ *is slope-restricted in the sector* $[\alpha, \beta]$. *Define* $\mathcal{T}_n$ *as in* (8). *Suppose there exists a* $\rho > 0$ *such that the matrix inequality*

$$M(\rho, T) := \begin{bmatrix} -2\alpha\beta W^{0^\top} T W^0 - \rho I_{n_0} & (\alpha+\beta)W^{0^\top} T \\ (\alpha+\beta)T W^0 & -2T + W^{1^\top} W^1 \end{bmatrix} \preceq 0, \tag{10}$$

*holds for some* $T \in \mathcal{T}_n$. *Then* $\|f(x) - f(y)\|_2 \leq \sqrt{\rho}\|x - y\|_2$ *for all* $x, y \in \mathbb{R}^{n_0}$.

Theorem 1 provides us with a sufficient condition for $L_2 = \sqrt{\rho}$ to be an upper bound on the Lipschitz constant of $f(x) = W^1\phi(W^0 x + b^0) + b^1$. In particular, we can find the tightest bound by solving the following optimization problem:

$$\text{minimize} \quad \rho \quad \text{subject to} \quad M(\rho, T) \preceq 0 \quad \text{and} \quad T \in \mathcal{T}_n, \tag{11}$$

where the decision variables are $(\rho, T) \in \mathbb{R}_+ \times \mathcal{T}_n$. Note that $M(\rho, T)$ is linear in $\rho$ and $T$ and the set $\mathcal{T}_n$ is convex. Hence, (11) is an SDP, which can be solved numerically for its global minimum.

### 2.4 LipSDP for multi-layer neural networks

We now consider the multi-layer case. Assuming that all the activation functions are the same, we can write the neural network model in (2) compactly as

$$B\mathbf{x} = \phi(A\mathbf{x} + b) \quad \text{and} \quad f(x) = C\mathbf{x} + b^\ell, \tag{12}$$

where $\mathbf{x} = [x^{0\top}\ x^{1\top} \cdots x^{\ell\top}]^\top$ is the concatenation of the input and the activation values, and the matrices $b$, $A$, $B$ and $C$ are given by [13]

$$A = \begin{bmatrix} W^0 & 0 & \dots & 0 & 0 \\ 0 & W^1 & \dots & 0 & 0 \\ \vdots & \vdots & \ddots & \vdots & \vdots \\ 0 & 0 & \dots & W^{\ell-1} & 0 \end{bmatrix}, \ B = \begin{bmatrix} 0 & I_{n_1} & 0 & \dots & 0 \\ 0 & 0 & I_{n_2} & \dots & 0 \\ \vdots & \vdots & \vdots & \ddots & \vdots \\ 0 & 0 & 0 & \dots & I_{n_\ell} \end{bmatrix}, \quad (13)$$

$$C = \begin{bmatrix} 0 & \dots & 0 & W^\ell \end{bmatrix}, \ b = \begin{bmatrix} b^{0\top} & \dots & b^{\ell-1\top} \end{bmatrix}^\top.$$

The particular representation in (12) facilitates the extension of `LipSDP` to multiple layers, as stated in the following theorem.

**Theorem 2 (Lipschitz certificates for multi-layer neural networks)** *Consider an $\ell$-layer fully connected neural network described by* (2). *Let $n = \sum_{k=1}^{\ell} n_k$ be the total number of hidden neurons and suppose the activation functions are slope-restricted in the sector $[\alpha, \beta]$. Define $\mathcal{T}_n$ as in* (8). *Define $A$ and $B$ as in* (13). *Consider the matrix inequality*

$$M(\rho, T) = \begin{bmatrix} A \\ B \end{bmatrix}^\top \begin{bmatrix} -2\alpha\beta T & (\alpha+\beta)T \\ (\alpha+\beta)T & -2T \end{bmatrix} \begin{bmatrix} A \\ B \end{bmatrix} + \begin{bmatrix} -\rho I_{n_0} & 0 & \dots & 0 \\ 0 & 0 & \dots & 0 \\ \vdots & \vdots & \ddots & \vdots \\ 0 & 0 & \dots & (W^\ell)^\top W^\ell \end{bmatrix} \preceq 0. \quad (14)$$

*If* (14) *is satisfied for some $(\rho, T) \in \mathbb{R}_+ \times \mathcal{T}_n$, then $\|f(x) - f(y)\|_2 \leq \sqrt{\rho}\,\|x - y\|_2$, $\forall x, y \in \mathbb{R}^{n_0}$.*

In a similar way to the single-layer case, we can find the best bound on the Lipschitz constant by solving the SDP in (11) with $M(\rho, T)$ defined as in (14).

**Remark 1** We have only considered the $\ell_2$ norm in our exposition. By using the inequality $\|x\|_p \leq n^{\frac{1}{p} - \frac{1}{q}} \|x\|_q$, the $\ell_2$-Lipschitz bound implies

$$n^{-(\frac{1}{p} - \frac{1}{2})} \|f(y) - f(x)\|_p \leq \|f(y) - f(x)\|_2 \leq L_2 \|y - x\|_2 \leq n^{\frac{1}{2} - \frac{1}{q}} L_2 \|y - x\|_q,$$

or, equivalently, $\|f(y) - f(x)\|_p \leq n^{\frac{1}{p} - \frac{1}{q}} L_2 \|y - x\|_q$. Hence, $n^{\frac{1}{p} - \frac{1}{q}} L_2$ is a Lipschitz constant of $f$ when $\ell_q$ and $\ell_p$ norms are used in the input and output spaces, respectively. We can also extend our framework to accommodate quadratic norms $\|x\|_P = \sqrt{x^\top P x}$, where $P \in \mathbb{S}_{++}^n$.

## 2.5 Variants of LipSDP: reconciling accuracy and efficiency

In `LipSDP`, there are $O(n^2)$ decision variables $\lambda_{ij}$, $1 \leq i, j \leq n$ ($\lambda_{ij} = \lambda_{ji}$), where $n$ is the total number of hidden neurons. For $i \neq j$, the variable $\lambda_{ij}$ couples the $i$-th and $j$-th hidden neuron. For $i = j$, the variable $\lambda_{ii}$ constrains the input-output of the $i$-th activation function individually. Using all these decision variables would provide the tightest convex relaxation in our formulation. However, solving this SDP with all the decision variables included is impractical for large networks. Nevertheless, we can consider a hierarchy of relaxations of `LipSDP` by removing a subset of the decision variables. Below, we give a brief description of the efficiency and accuracy of each variant. Throughout, we let $n$ be the total number of neurons and $\ell$ the number of hidden layers.

1. **LipSDP-Network** imposes constraints on all possible pairs of activation functions and has $O(n^2)$ decision variables. It is the least scalable but the most accurate method.

2. **LipSDP-Neuron** ignores the cross coupling constraints among different neurons and has $O(n)$ decision variables. It is more scalable and less accurate than LipSDP-Network. For this case, we have $T = \mathrm{diag}(\lambda_{11}, \cdots, \lambda_{nn})$.

3. **LipSDP-Layer** considers only one constraint per layer, resulting in $O(\ell)$ decision variables. It is the most scalable and least accurate method. For this variant, we have $T = \mathrm{blkdiag}(\lambda_1 I_{n_1}, \cdots, \lambda_\ell I_{n_\ell})$.

**Parallel implementation by splitting**. The Lipschitz constant of the composition of two or more functions can be bounded by the product of the Lipschitz constants of the individual functions. By

| $n$ | LipSDP-Neuron | LipSDP-Layer |
|-----|---------------|--------------|
| 500 | 5.22 | 2.85 |
| 1000 | 27.91 | 17.88 |
| 1500 | 82.12 | 58.61 |
| 2000 | 200.88 | 146.09 |
| 2500 | 376.07 | 245.94 |
| 3000 | 734.63 | 473.25 |

**Table 1:** Computation time in seconds for evaluating Lipschitz bounds of one-hidden-layer neural networks with a varying number of hidden units. A plot showing the Lipschitz constant for each network tested in this table has been provided in the Appendix.

| $\ell$ | LipSDP-Neuron | LipSDP-Layer |
|--------|---------------|--------------|
| 5 | 20.33 | 3.41 |
| 10 | 32.18 | 7.06 |
| 50 | 87.45 | 25.88 |
| 100 | 135.85 | 40.39 |
| 200 | 221.2 | 64.90 |
| 500 | 707.56 | 216.49 |

**Table 2:** Computation time in seconds for computing Lipschitz bounds of $\ell$-hidden-layer neural networks with 100 activation functions per layer. For `LipSDP-Neuron` and `LipSDP-Layer`, we split each network up into 5-layer sub-networks.

splitting a neural network up into small sub-networks, one can first bound the Lipschitz constant of each sub-network and then multiply these constants together to obtain a Lipschitz constant for the entire network. Because sub-networks do not share weights, it is possible to compute the Lipschitz constants for each sub-network in parallel. This greatly improves the scalability of each variant of `LipSDP` with respect to the total number of activation functions in the network. We remark that this parallelization is not exclusive to our method. However, among all methods that can split the computations across layers, our method yields more accurate bounds per split.

## 3 Experiments

In this section we describe several experiments that highlight the key aspects of this work. In particular, we show empirically that our bounds are much tighter than any comparable method, we study the impact of robust training on our Lipschitz bounds, and we analyze the scalability of our methods.

**Experimental setup.** For our experiments we used MATLAB, the CVX toolbox [18] and MOSEK [3] on a 9-core CPU with 16GB of RAM to solve the SDPs. All classifiers trained on MNIST used an 80-20 train-test split.

**Training procedures.** Several training procedures have recently been proposed to improve the robustness of neural network classifiers. Two prominent procedures are the LP-based method in [40] and projected gradient descent (PGD) based method in [24]. We refer to these training methods as `LP-Train` and `PGD-Train`, respectively. Both procedures take as input a parameter $\epsilon$ that defines the $\ell_\infty$ perturbation of the training data points.

**Baselines.** Throughout the experiments, we will often show comparisons to the naive upper bound on the Lipschitz constant given by $L_{2,\,\text{upper}} = \prod_{i=0}^{\ell} \left|\left|W^i\right|\right|_2$. We are aware of only two methods that bound the Lipschitz constant and can scale to fully-connected networks with more than two hidden layers; these methods are [10], which we will refer to as `CPLip`, and [37], which is called `SeqLip`. We compare the Lipschitz bounds obtained by `LipSDP-Neuron`, `LipSDP-Layer`, `CPLip`, and `SeqLip` in Figure 2a. It is evident from this figure that the bounds from `LipSDP-Neuron` are tighter than `CPLip` and `SeqLip`.

To demonstrate the scalability of the `LipSDP` formulations, we split a 100-hidden layer neural network into sub-networks with six hidden layers each and computed the Lipschitz bounds using `LipSDP-Neuron` and `LipSDP-Layer`. The results are shown in Figure 2b. Furthermore, in Tables 1 and 2, we show the computation time for scaling the `LipSDP` methods in the number of hidden units per layer and in the number of layers. In particular, the largest network we tested in Table 2 had 50,000 hidden neurons; `SDPLip-Neuron` took approximately 12 minutes to find a Lipschitz bound, and `SDPLip-Layer` took approximately 4 minutes.

To evaluate `SDPLip-Network`, we coupled random pairs of hidden neurons in a one-hidden-layer network and plotted the computation time and Lipschitz bound found by `SDPLip-Network` as we increased the number of paired neurons. Our results show that as the number of coupled neurons

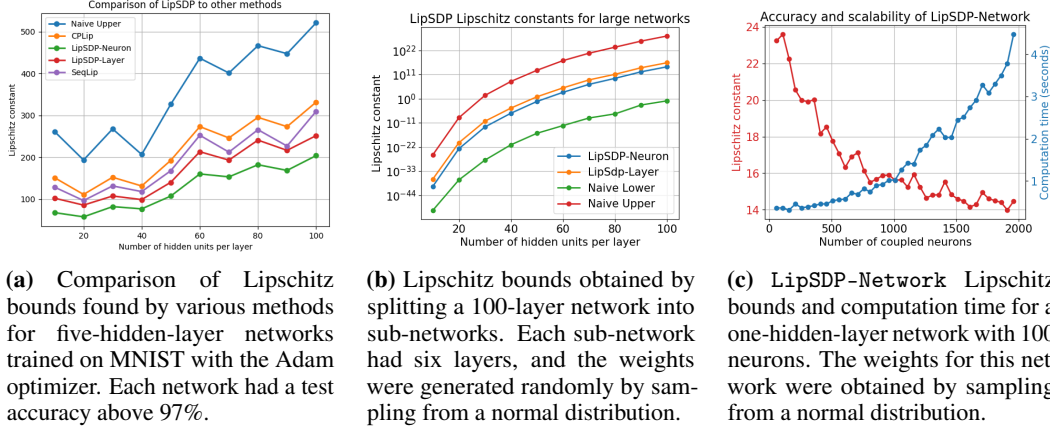

**(a)** Comparison of Lipschitz bounds found by various methods for five-hidden-layer networks trained on MNIST with the Adam optimizer. Each network had a test accuracy above 97%.

**(b)** Lipschitz bounds obtained by splitting a 100-layer network into sub-networks. Each sub-network had six layers, and the weights were generated randomly by sampling from a normal distribution.

**(c)** `LipSDP-Network` Lipschitz bounds and computation time for a one-hidden-layer network with 100 neurons. The weights for this network were obtained by sampling from a normal distribution.

**Figure 2:** Comparison of the accuracy `LipSDP` methods to other methods that compute the Lipschitz constant and scalability analysis of all three `SeqLip` methods.

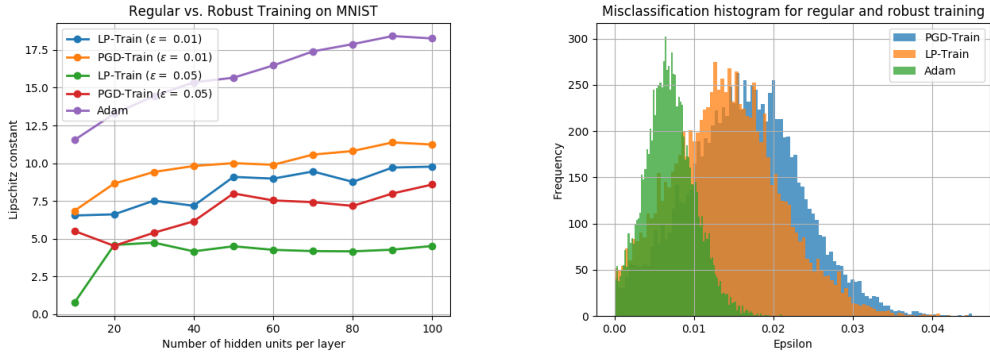

**(a)** Lipschitz bounds for a one-hidden-layer neural networks trained on the MNIST dataset with the Adam optimizer and `LP-Train` and `PGD-Train` for two values of the robustness parameter $\epsilon$. Each network reached an accuracy of 95% or higher.

**(b)** Histograms showing the local robustness (in $\ell_\infty$ norm) around each correctly-classified test instance from the MNIST dataset. The neural networks had three hidden layers with 100, 50, 20 neurons, respectively. All classifiers had a test accuracy of 97%. We used Remark 1 to convert the norm from $\ell_2$ to $\ell_\infty$.

**Figure 3:** Analysis of impact of robust training on the Lipschitz constant and the distance to misclassification for networks trained on MNIST

increases, the computation time increases quadratically. This shows that while this method is the most accurate of the three proposed `LipSDP` methods, it is intractable for even modestly large networks.

**Impact of robust training.** In Figure 3, we empirically demonstrate that the Lipschitz bound of a neural network is directly related to the robustness of the corresponding classifier. This figure shows that `LP-train` and `PGD-Train` networks achieve lower Lipschitz bounds than standard training procedures. Figure 3a indicates that robust training procedures yield lower Lipschitz constants than networks trained with standard training procedures such as the Adam optimizer. Figure 3b shows the utility of sharply estimating the Lipschitz constant; a lower value of $L_2$ guarantees that a neural network is more locally robust to input perturbations; see Proposition 1 in the Appendix.

In the same vein, Figure 4 shows the impact of varying the robustness parameter $\epsilon$ used in `LP-Train` and `PGD-Train` on the test accuracy of networks trained for a fixed number of epochs and the corresponding Lipschitz constants. In essence, these results quantify how much robustness a fixed classifier can handle before accuracy plummets. Interestingly, the drops in accuracy as $\epsilon$ increases coincide with corresponding drops in the Lipschitz constant for both `LP-Train` and `PGD-Train`.

**Robustness for different activation functions.** The framework proposed in this work allows us to examine the impact of using different activation functions on the Lipschitz constant of neural

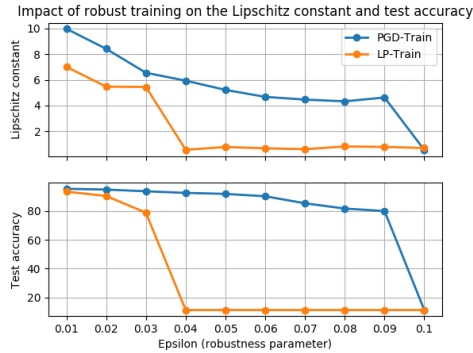

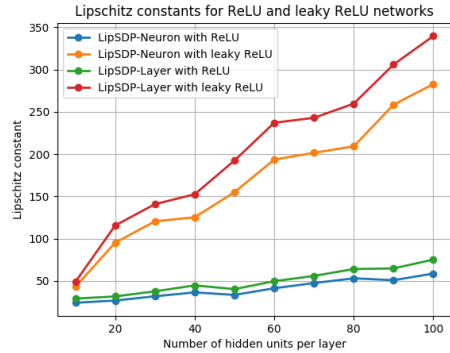

**Figure 4:** Trade-off between accuracy and Lipschitz constant for different values of the robustness parameter used for `LP-Train` and `PGD-Train`. All networks had one hidden layer with 50 hidden neurons.

**Figure 5:** Lipschitz constants for topologically identical three-hidden-layer networks with ReLU and leaky ReLU activation functions. All classifiers were trained until they reached 97% test accuracy.

networks. We trained two sets of neural networks on the MNIST dataset. The first set used ReLU activation functions, while the second set used leaky ReLU activations. Figure 5 shows empirically that the networks with the leaky ReLU activation function have larger Lipschitz constants than networks of the same architecture with the ReLU activation function.

## 4   Conclusions and future work

In this paper, we proposed a hierarchy of semidefinite programs to derive tight upper bounds on the Lipschitz constant of feed-forward fully-connected neural networks. Some comments are in order. First, our framework can be directly used to certify convolutional neural networks (CNNs) by unrolling them to a large fully-connected neural network. This conversion implicitly handles the padding and stride hyper parameters. Since the max function is convex, we can describe the max pooling operation using incremental quadratic constraints without additional assumptions. Therefore, in principle, `LipSDP` is applicable to CNNs. A future direction is to exploit the special structure of CNNs in the resulting SDP. Second, we only considered one application of Lipschitz bounds in depth (robustness certification). Having an accurate upper bound on the Lipschitz constant can be useful in domains beyond robustness analysis, such as stability analysis of feedback systems with control policies updated by deep reinforcement learning. Furthermore, Lipschitz bounds can be utilized during training as a heuristic to promote out-of-sample generalization [36]. We intend to pursue these applications for future work.

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
