[Supplementary Material]

This is the supplementary material for the NeurIPS 2019 paper: "Efficient and Accurate Estimation of Lipschitz Constants for Deep Neural Networks", by Mahyar Fazlyab, Alexander Robey, Hamed Hassani, Manfred Morari, and George J. Pappas.

## A Appendix

### A.1 Robustness certification of DNN-based classifiers

Consider a classifier described by a feed-forward neural network $f \colon \mathbb{R}^n \to \mathbb{R}^k$, where $n$ is the number of input features and $k$ is the number of classes. In this context, the function $f$ takes as input an instance or measurement $x$ and returns a $k$-dimensional vector of scores – one for each class. The classification rule is based on assigning $x$ to the class with the highest score. That is, we define the classification $C(x) \colon \mathbb{R}^n \to \{1, \ldots, k\}$ to be $C(x) = \operatorname{argmax}_{1 \le i \le k} f_i(x)$. Now suppose that $x^\star$ is an instance that is classified correctly by the neural network. To evaluate the local robustness of the neural network around $x^\star$, we consider a bounded set $\mathcal{A}(x^\star) = \{x \mid \|x - x^\star\|_2 \le \epsilon\} \subseteq \mathbb{R}^n$ that represents the set of all possible $\ell_2$-norm perturbations of $x^\star$. Then the classifier is locally robust at $x^\star$ against $\mathcal{A}(x^\star)$ if it assigns all the perturbed inputs to the same class as the unperturbed input, i.e., if

$$C(x) = C(x^\star) \quad \forall x \in \mathcal{A}(x^\star). \tag{15}$$

In the following proposition, we derive a sufficient condition to guarantee local robustness around $x^\star$ for the perturbation set $\mathcal{A}(x^\star)$.

**Proposition 1** *Consider a neural-network classifier $f \colon \mathbb{R}^n \to \mathbb{R}^k$ with Lipschitz constant $L_2$ in the $\ell_2$-norm. Let $\epsilon > 0$ be given and consider the inequality*

$$L_2 \le \frac{1}{\epsilon\sqrt{2}} \min_{1 \le j \le k,\ j \ne i^\star} |f_j(x^\star) - f_{i^\star}(x^\star)|. \tag{16}$$

*Then* (16) *implies* $C(x) := \operatorname{argmax}_{1 \le i \le k} f_i(x) = C(x^\star)$ *for all* $x \in \mathcal{A}(x^\star)$.

The inequality in (16) provides us with a simple and computationally efficient test for assessing the point-wise robustness of a neural network. According to (16), a more accurate estimation of the Lipschitz constant directly increases the maximum perturbation $\epsilon$ that can be certified for each test example. This makes the framework suitable for model selection, wherein one wishes to select the model that is most robust to adversarial perturbations from a family of proposed classifiers [27].

### A.2 Proof of Proposition 1

Let $i^\star = \operatorname{argmax}_{1 \le i \le k} f_i(x^\star)$ be the class of $x^\star$. Define the polytope in the output space of $f$:

$$\mathcal{P}_{i^\star} = \{y \in \mathbb{R}^k \mid (e_j - e_{i^\star})^\top y \le 0 \quad 1 \le j \le k,\ j \ne i^\star\}.$$

which is the set of all outputs whose score is the highest for class $i^\star$; and, of course, $f(x^\star) \in \mathcal{P}_{i^\star}$. The distance of $f(x^\star)$ to the boundary $\partial\mathcal{P}_i$ of the polytope is the minimum distance of $f(x^\star)$ to all edges of the polytope:

$$\mathbf{dist}(f(x^\star), \partial\mathcal{P}_{i^\star}) = \inf_{y \in \partial\mathcal{P}_{i^\star}} \|y - f(x^\star)\|_2 = \frac{1}{\sqrt{2}} \min_{1 \le j \le k,\ j \ne i^\star} |f_j(x^\star) - f_{i^\star}(x^\star)|.$$

Note that the Lipschitz condition implies $\mathbf{dist}(f(x^\star), f(x)) = \|f(x) - f(x^\star)\|_2 \le L_2\epsilon$ for all $\|x - x^\star\|_2 \le \epsilon$. The condition in (16) then implies

$$\mathbf{dist}(f(x^\star), f(x)) \le \mathbf{dist}(f(x^\star), \partial\mathcal{P}_{i^\star}) \ \text{ for all } \|x - x^\star\|_2 \le \epsilon,$$

and for all $1 \le j \le k,\ j \ne i^\star$. Therefore, the output of the classification would not change for any $\|x - x^\star\|_2 \le \epsilon$.

### A.3 Proof of Lemma 1

We first prove the following lemma, which is a slight variation to the lemma proved in [13].

**Figure 6:** Illustration of local robustness certification using the Lipschitz bound.

**Lemma 2** *Suppose $\varphi \colon \mathbb{R} \to \mathbb{R}$ is slope-restricted on $[\alpha, \beta]$ and satisfies $\varphi(0) = 0$. Define the set*

$$\mathcal{T}_n = \{T \in \mathbb{S}^n \mid T = \sum_{i=1}^{n} \lambda_{ii} e_i e_i^\top + \sum_{1 \leq i < j \leq n} \lambda_{ij}(e_i - e_j)(e_i - e_j)^\top, \lambda_{ij} \geq 0\}. \qquad (17)$$

*Then the vector-valued function $\phi : \mathbb{R}^n \to \mathbb{R}^n$ defined by $\phi(x) = [\varphi(x_1) \cdots \varphi(x_n)]^\top$ satisfies*

$$\begin{bmatrix} x \\ \phi(x) \end{bmatrix}^\top \begin{bmatrix} -2\alpha\beta T & (\alpha + \beta)T \\ (\alpha + \beta)T & -2T \end{bmatrix} \begin{bmatrix} x \\ \phi(x) \end{bmatrix} \geq 0 \quad \forall x \in \mathbb{R}^n, \qquad (18)$$

*for all $T \in \mathcal{T}_n$.*

**Proof of Lemma 2**. Note that the slope restriction condition implies that for any two pairs $(x_i, \varphi(x_i))$ and $(x_j, \varphi(x_j))$, we can write the following incremental quadratic constraint:

$$\begin{bmatrix} x_i - x_j \\ \varphi(x_i) - \varphi(x_j) \end{bmatrix}^\top \left( \begin{bmatrix} -2\alpha\beta & \alpha + \beta \\ \alpha + \beta & -2 \end{bmatrix} \cdot \lambda_{ij} \right) \begin{bmatrix} x_i - x_j \\ \varphi(x_i) - \varphi(x_j) \end{bmatrix} \geq 0 \quad 1 \leq i \neq j \leq n. \qquad (19)$$

where $\lambda_{ij} \geq 0$ is arbitrary. Similarly, for any two pairs $(x_i, \varphi(x_i))$ and $(0, \varphi(0)) = (0, 0)$, we can also write the incremental quadratic constraint

$$\begin{bmatrix} x_i - 0 \\ \varphi(x_i) - 0 \end{bmatrix}^\top \left( \begin{bmatrix} -2\alpha\beta & \alpha + \beta \\ \alpha + \beta & -2 \end{bmatrix} \cdot \lambda_{ii} \right) \begin{bmatrix} x_i - 0 \\ \varphi(x_i) - 0 \end{bmatrix} \geq 0 \quad i = 1, \cdots, n. \qquad (20)$$

where $\lambda_{ii} \geq 0$ is arbitrary. By adding (19) and (20) and vectorizing the notation, we would arrive at the compact representation (18).

**Proof of Lemma 1.** For a fixed $z \in \mathbb{R}$, define the map $\tilde{\varphi}$ by shifting $\varphi$, as follows,

$$\tilde{\varphi}(z, \delta) = \varphi(z + \delta) - \varphi(z) \quad \delta \in \mathbb{R}.$$

It is not hard to verify that if $\varphi$ is slope-restricted on $[\alpha, \beta]$, the map $\delta \mapsto \tilde{\varphi}(z, \delta)$ is also slope-restricted on the same interval for any fixed $z \in \mathbb{R}$. Next, for a fixed $x \in \mathbb{R}^n$ define

$$\tilde{\phi}(x, \delta) = \phi(x + \delta) - \phi(x) = [\tilde{\varphi}(x_1, \delta_1) \cdots \tilde{\varphi}(x_n, \delta_n)]^\top \quad \delta \in \mathbb{R}^n.$$

Since $\tilde{\varphi}$ is slope-restricted on $[\alpha, \beta]$ and satisfies $\tilde{\varphi}(z, 0) = 0$ for any fixed $z \in \mathbb{R}$, it follows from Lemma 2 that $\tilde{\phi}(x, \delta)$ satisfies the quadratic constraint

$$\begin{bmatrix} \delta \\ \tilde{\phi}(x, \delta) \end{bmatrix}^\top \begin{bmatrix} -2\alpha\beta T & (\alpha + \beta)T \\ (\alpha + \beta)T & -2T \end{bmatrix} \begin{bmatrix} \delta \\ \tilde{\phi}(x, \delta) \end{bmatrix} \geq 0.$$

By substituting the definition of $\tilde{\phi}(x, \delta)$ and setting $\delta = y - x$, we obtain

$$\begin{bmatrix} y - x \\ \phi(y) - \phi(x) \end{bmatrix}^\top \begin{bmatrix} -2\alpha\beta T & (\alpha + \beta)T \\ (\alpha + \beta)T & -2T \end{bmatrix} \begin{bmatrix} y - x \\ \phi(y) - \phi(x) \end{bmatrix} \geq 0. \qquad (21)$$

## A.4 Proof of Theorem 1

Define $x^1 = \phi(W^0 x + b^0) \in \mathbb{R}^n$ and $y^1 = \phi(W^0 y + b^0) \in \mathbb{R}^n$ for two arbitrary inputs $x, y \in \mathbb{R}^{n_0}$. Using Lemma 1, we can write the quadratic inequality

$$0 \leq \begin{bmatrix} (W^0 x + b^0) - (W^0 y + b^0) \\ x^1 - y^1 \end{bmatrix}^\top \begin{bmatrix} -2\alpha\beta T & (\alpha+\beta)T \\ (\alpha+\beta)T & -2T \end{bmatrix} \begin{bmatrix} (W^0 x + b^0) - (W^0 y + b^0) \\ x^1 - y^1 \end{bmatrix},$$

where $T \in \mathcal{T}_n$ and $\mathcal{T}_n$ is defined as in (8). The preceding inequality can be simplified to

$$0 \leq \begin{bmatrix} x-y \\ x^1-y^1 \end{bmatrix}^\top \begin{bmatrix} -2\alpha\beta W^{0\top} T W^0 & (\alpha+\beta)W^{0\top}T \\ (\alpha+\beta)TW^0 & -2T \end{bmatrix} \begin{bmatrix} x-y \\ x^1-y^1 \end{bmatrix}. \tag{22}$$

By left and right multiplying $M(\rho, T)$ in (10) by $\begin{bmatrix} (x-y)^\top & (x^1-y^1)^\top \end{bmatrix}$ and $\begin{bmatrix} (x-y)^\top & (x^1-y^1)^\top \end{bmatrix}^\top$, respectively, and rearranging terms, we obtain

$$\begin{bmatrix} x-y \\ x^1-y^1 \end{bmatrix}^\top \begin{bmatrix} -2\alpha\beta W^{0\top} T W^0 & (\alpha+\beta)W^{0\top}T \\ (\alpha+\beta)TW^0 & -2T \end{bmatrix} \begin{bmatrix} x-y \\ x^1-y^1 \end{bmatrix} \tag{23}$$

$$\leq \begin{bmatrix} x-y \\ x^1-y^1 \end{bmatrix}^\top \begin{bmatrix} L_2^2 I_{n_0} & 0 \\ 0 & -W^{1\top}W^1 \end{bmatrix} \begin{bmatrix} x-y \\ x^1-y^1 \end{bmatrix}.$$

By adding both sides of the preceding inequalities, we obtain

$$0 \leq \begin{bmatrix} x-y \\ x^1-y^1 \end{bmatrix}^\top \begin{bmatrix} L_2^2 I_{n_0} & 0 \\ 0 & -W^{1\top}W^1 \end{bmatrix} \begin{bmatrix} x-y \\ x^1-y^1 \end{bmatrix},$$

or, equivalently,

$$(x^1 - y^1)^\top W^{1\top} W^1 (x^1 - y^1) \leq L_2^2 (x-y)^\top (x-y).$$

Finally, note that by definition of $x^1$ and $y^1$, we have $f(x) = W^1 x^1 + b^1$ and $f(y) = W^1 y^1 + b^1$. Therefore, the preceding inequality implies

$$\|f(x) - f(y)\|_2 \leq L_2 \|x - y\|_2 \quad \text{for all } x, y \in \mathbb{R}^{n_0}. \tag{24}$$

## A.5 Proof of Theorem 2

For two arbitrary inputs $x^0, y^0 \in \mathbb{R}^{n_0}$, define $\mathbf{x} = [x^{0\top} \cdots x^{\ell\top}]^\top$ and $\mathbf{y} = [y^{0\top} \cdots y^{\ell\top}]^\top$ Using the compact notation in (12), we can write

$$B\mathbf{x} = \phi(A\mathbf{x} + b) \quad \text{and} \quad B\mathbf{y} = \phi(A\mathbf{y} + b).$$

Multiply both sides of the first matrix in (14) by $(\mathbf{x} - \mathbf{y})^\top$ and $(\mathbf{x} - \mathbf{y})$, respectively and use the preceding identities to obtain

$$(\mathbf{x} - \mathbf{y})^\top \begin{bmatrix} A \\ B \end{bmatrix}^\top \begin{bmatrix} -2\alpha\beta T & (\alpha+\beta)T \\ (\alpha+\beta)T & -2T \end{bmatrix} \begin{bmatrix} A \\ B \end{bmatrix} (\mathbf{x} - \mathbf{y}) \tag{25}$$

$$= \begin{bmatrix} A\mathbf{x} - A\mathbf{y} \\ B\mathbf{x} - B\mathbf{y} \end{bmatrix}^\top \begin{bmatrix} -2\alpha\beta T & (\alpha+\beta)T \\ (\alpha+\beta)T & -2T \end{bmatrix} \begin{bmatrix} A\mathbf{x} - A\mathbf{y} \\ B\mathbf{x} - B\mathbf{y} \end{bmatrix}$$

$$= \begin{bmatrix} A\mathbf{x} - A\mathbf{y} \\ \phi(A\mathbf{x} + b) - \phi(A\mathbf{y} + b) \end{bmatrix}^\top \begin{bmatrix} -2\alpha\beta T & (\alpha+\beta)T \\ (\alpha+\beta)T & -2T \end{bmatrix} \begin{bmatrix} A\mathbf{x} - A\mathbf{y} \\ \phi(A\mathbf{x} + b) - \phi(A\mathbf{y} + b) \end{bmatrix} \geq 0,$$

where the last inequality follows from Lemma 1. On the other hand, by multiplying both sides of the second matrix in (14) by $(\mathbf{x} - \mathbf{y})^\top$ and $(\mathbf{x} - \mathbf{y})$, respectively, we can write

$$(\mathbf{x} - \mathbf{y})^\top M (\mathbf{x} - \mathbf{y}) = \|f(x) - f(y)\|_2^2 - L_2^2 \|x - y\|_2^2, \tag{26}$$

where we have used the fact that $f(x) = W^\ell x^\ell + b^\ell$ and $f(y) = W^\ell y^\ell + b^\ell$. By adding both sides of (25) and (26), we get

$$(\mathbf{x} - \mathbf{y})^\top \left( \begin{bmatrix} A \\ B \end{bmatrix}^\top \begin{bmatrix} -2\alpha\beta T & (\alpha+\beta)T \\ (\alpha+\beta)T & -2T \end{bmatrix} \begin{bmatrix} A \\ B \end{bmatrix} + M \right) (\mathbf{x} - \mathbf{y}) \geq \|f(x) - f(y)\|_2^2 - L_2^2 \|x - y\|_2. \tag{27}$$

When the LMI in (14) holds, the left-hand side of (27) is non-positive, implying that the right-hand side is non-positive.

**Figure 7:** Bounds on the image of an $\ell_2$ 0.01-ball under a neural network trained on the Iris dataset. The network was trained using the Adam optimizer and reached a test accuracy of 99%.

**Figure 8:** Lipschitz bounds for networks in Table 1.

**Figure 9:** Test accuracy and bounds on the Lipschitz constant for neural networks trained on MNIST using the training method in [17].

## A.6   Bounding the output set of a neural network classifier

As we have shown, accurate estimation of the Lipschitz constant can provide tighter bounds on adversarial examples drawn from a perturbation set around a nominal point $\bar{x}$. Figure 7 shows these bounds for a neural network trained on the Iris dataset. Because this dataset has three classes, we project the output sets onto the coordinate axes.

## A.7   Further analysis of scalability

Figure 8 shows the Lipschitz bounds found for the networks used in Table 1.

## A.8   Additional experiments

In [17] the authors formulate training a neural network with a bounded Lipschitz constant as a constrained optimization problem that can be solved using projected stochastic gradient descent. We used this method to train a feed-forward fully-connected neural network with architecture (784-200-100-10) on MNIST for various values of $\lambda$ – a hyper-parameter constraining the Lipschitz constant of each layer. In Figure 9, we plot the test accuracy and the Lipschitz bounds as a function of $\lambda$. As the bound on the Lipschitz constant of each layer increases (i.e., $\lambda$), the resulting network achieves a higher test accuracy and a larger Lipschitz constant. Furthermore, the bounds obtained from `LipSDP` are more accurate than the bounds guaranteed by the training method.