[Reviews · NeurIPS 2019]

Reviewer 1



Originality? This work makes new observations and exploits intimate knowledge of convex optimization unseen in prior work I'm aware of. Quality? The claims are theoretically justified and experimentally validated in small models. Clarity? The submission is clearly written and easy to understand. Significance? This work solves the problem of Lipschitz constant estimation of DNNs in a convincingly superior way to prior work. The output of LipSDP comes with an upper-bound certificate and returns accurate approximations in the proposed experiments. I believe this work adds great insight and advances the SOTA in its area. The primary limitation is that the method currently only applies to small, feed-forward fully-connected networks.

Reviewer 2



*** Comments after feedback *** thanks for the clarifications. with the proposed minor improvements I feel the paper is solid so I will raise to 8. However I stress it would be good to remark that the parallelisation scheme is not exclusive to your method. *** Original review *** Originality. Even though other optimization-based certification of Lipschitz constants have been proposed before, the theoretical result leading to the SDP formulation (Theorem 1) is novel. The multiple variants of the main algorithm aim to provide a more scalable method and in part succeeds at doing so (evaluated networks are still relatively simple), and despite the loss in accuracy the less-complex version of the methodology can achieve better or competitive estimation of the constant. On the other hand the authors overplay the fact that any method that estimates a Lipschitz constant on a multilayer network can be trivially parallelized by splitting the network into "chunks". This is not a particular advantage of their method and so I think the claims about the parallel version should be toned down. Even from the trivial upper bound on the Lipschitz constant, given by the product of the layer-wise constants, it is clear that such methods can be easily parallelized. Quality. The proofs are well written and after checking the supplementary material I found no issues on the results leading to theorem 1 (theorem 2 looks like a simple extension of theorem 1 and so I didn't check). The derived bounds are then theoretically justified. On the other hand I find there is a mismatch between theory and practice in the experimental section. Even though the method certifies a Lipschitz constant with respect to the L2 metric, the adversarial perturbations studied are constrained with the L_infinity metric. So the authors should explain or remark that this is the case. Ideally if we certify an L2 Lipschitz constant, this would provide robustness w.r.t the L2 metric. This is a general drawback of the work where there is no discussion as to how to compute the Lipschitz constant with respect to other norms (apart from the trivial bounds), or if there is a general limitation of the method that allows it to only work with the L2 metric. Authors should comment a bit more on this. Clarity. The paper was read easily, and statements are in general clear. On the miscellaneous side I feel the bibliography style is not consistent, it is typeset with [1][2][3] style of citations but sometimes also the names of authors are present e.g. page 2 third paragraph: "in [30], Virmaux and Scaman ...". I found this confusing. Also there appears to be a typo following theorem 1: "Theorem 1 provides us with a sufficient condition for the constant L2..." I think it should read "for the constant \sqrt{\rho}..." Significance. Accurate estimation of Lipschitz constant has a broad range of applications, and the empirical observation that the obtained constants are tighter compared to other methods make the contribution significant. On the other hand the underlying problem is an SDP which is still difficult to scale, and the fact that the experiments are only presented on MNIST makes one wonder how would such methods work with larger and more complex network architectures.

Reviewer 3



1) The submitted paper is in violation of the style requirements. For example, there are no line numbers. 2) I found the notation as presented in 2.1 quite unclear. I believe that adopting bold-face for vector-valued quantities would make things much easier to parse --- right now these are differentiated by sub/super-script. I believe that the function phi us the concatenation of activation functions for each neuron in a layer. The current wording sounds as though this function concatenates all the layers outputs together. 3) The language used around the assumptions on activation functions varies. In some places the authors write that all activation functions satisfy the required assumptions but in others note that all common activation functions are admissible. The latter is more reasonable and should be used throughout (I do not think this weakens the paper, but improves clarity). In particular, the first sentence in the "Our approach" section and following Definition 1: "Note that all activation functions used in deep learning [...]". As an example of why I think this should be done carefully, see point (6c) where an activation function which (I think) violates the assumptions has been used for Lipschitz neural networks. 4) I found Figure 1 difficult to understand. Minor improvements could be achieved by fixing spelling errors ('combintation'), making ordering consistent (yi-xi in equation but y > x in image), and improving general visual clarity. Notation could also be made consistent to help the flow of information (e.g. y_i and y_j appear in the second block, but nowhere else). Further to this, the figure caption serves no real purpose and does not help an uninformed reader understand the derivation of the quadratic constraints. 5) I thought that the proposed method was a little hard for someone to understand without a strong background in convex programming, but I like the method in general. If I understand correctly, the method should be easy to scale to large fully-connected architectures. One potential draw back of the method is the assumption that the architectures are fully-connected. The authors note in the conclusion that this may be extended to convolutional neural networks but I am not certain that this can be done seamlessly as there are typically some subtleties in achieving tight bounds under arbitrary input-sizes/strides/padding/etc. [1]. 6) I think it might be worthwhile to acknowledge the relatively small body of work which explores universality under Lipschitz constraints [e.g. 2,3]. The goal of this work is not to estimate the Lipschitz constant tightly but rather to train networks under a given Lipschitz constraint well. This is related in two ways. First, the methods used to achieve this enable easy computation of the Lipschitz constant of the network. Second, [2] highlights the inability of most activation functions to produce networks that achieve the specified Lipschitz constant which is related to the gap between the naive upper bound and the true Lipschitz constant --- this may provide a good setting to evaluate LipSDP. 6b) [2] also proposes an alternative scalable approach to estimating the Lipschitz constant which is not discussed here (but likely appears in other work). The maximum spectral norm of the network Jacobian (taken over the data distribution) is used as an estimate of the true Lipschitz constant. This will almost certainly underestimate the true Lipschitz constant which speaks further to the advantage of LipSDP. 6c) Finally, the main results of [2] depend on an activation function which violated the necessary assumptions of Theorems 1 and 2 (I believe). 7) I thought the experiments were fairly interesting but limited. It would be good to compare LipSDP to other methods on more than one dataset and to provide error bars over different training runs. I thought it was interesting to explore how adversarial training (and other robustness mechanisms) may change the Lipschitz constant of the network and would have liked to have seen some expansion on this (either empirically or theoretically). I think that it would also be interesting to train networks under a given Lipschitz constraint and compare this to the output of LipSDP (e.g. methods proposed in [1,2,3], and others). 8) I don't understand exactly how the naive lower bound is indeed a lower bound. For a deep linear network this provides the correct Lipschitz constant. Introducing non-linear activation functions with Lipschitz constant strictly less than 1 will thus decrease the Lipschitz constant below the naive lower bound. Am I missing something here? Could you please explain under which assumptions this quantity provides a lower bound? (I was unable to find this from a quick scan of the provided references). Summary: I like the proposed method in this paper and consider this a significant contribution. I think that the empirical results could have been improved but I do think they provide adequate evidence that LipSDP offers advantages over existing approaches. Minor comments: - Passive citations (citep) used actively in places (e.g. Section 3, Baselines.) Clarity: I felt that technical detail and figures could be improved to make the proposed method easier to understand. Overall, the clarity of the paper was moderate. Significance: This paper provides an alternative method to estimate the Lipschitz constant of neural networks and does so more tightly while scaling well. This work builds on previously presented ideas but enables wider ranging applications. Originality: To my knowledge the proposed method is novel and the work offers new empirical insights surrounding robust training methods and Lipschitz constraints. References: [1] "Regularisation of Neural Networks by Enforcing Lipschitz Continuity", Gouk et al. [2] "Sorting out Lipschitz function approximation", Anil et al. [3] "Limitations of the Lipschitz constant as a defense against adversarial examples", Huster et al.

[Author Response · NeurIPS 2019]

**Authors' Response for NeurIPS Paper Number 6086.** We thank the reviewers for their constructive comments. We first provide a general response common to all the reviewers and then respond to each reviewer's particular comments.

**Beyond MNIST and Feed-Forward Fully-Connected (FFC):** Since the submission of the paper, we have moved beyond MNIST and FFC neural networks; in particular, we have performed experiments on several CNNs trained on CIFAR-10 and confirmed that (a) our method can scale to larger and more complex networks*, and (b) our bounds are consistently superior to the competing approaches. We will release the code necessary to reproduce these results. In its current status our approach scales to non-trivial tasks such as deep RL for control or robotics [5,6,7]. We can further improve the scalability of our tool by simply improving our hardware (computations were performed on a basic laptop) and our software (exploiting the structure in the SDP, for example).

*We investigated CNNs with two convolutional layers (with three and six filters, no padding, and a stride of one) followed by one linear layer. We remark that unrolling the convolutional operators in a CNN results in a (large) feedforward network. This conversion implicitly handles the padding and stride hyperparameters. Furthermore, since the max function is convex, we can describe the max pooling operation using quadratic constraints without additional assumptions. Therefore, we can directly use `LipSDP` for CNNs. The CNNs we used had approximately 15,000 neurons after unrolling.

**Reviewer 1:** *Software release:* We have already implemented the software necessary to reproduce our results. All of the code is included in a private GitHub repository that will be made public with the camera-ready version of this paper. Our software links to both interior-point method solvers (Mosek) and first-order method solvers, which are specifically tailored for large-scale cone programs. We tried several large-scale solvers but found the SCS solver, developed by Boyd's group, to be the most stable one.

**Reviewer 2:** *Lipschitz constant w.r.t. other norms:* In our experiments, we bounded the Lipschitz constant in $\ell_2$, then derived the adversarial balls in $\ell_2$, and finally converted the $\ell_2$ balls to $\ell_\infty$ balls using the inequality $\|x\|_p \leq n^{\frac{1}{p} - \frac{1}{q}} \|x\|_q$ ($x \in \mathbb{R}^n$). In Remark 1, we discussed how to convert the Lipschitz bound from $\ell_2$ norm to other $\ell_p$-norms. Apart from this norm conversion, we believe that reformulating the problem (by, e.g., moving to the dual domain or changing the objective function) will enable us to incorporate other $\ell_p$ norms ($p = 1, 2, \infty$) directly in `LipSDP`. A thorough treatment of this is an important future research direction. *Parallelization:* In principle, we agree with the reviewer that the splitting+parallelization scheme discussed in our paper is not exclusive to our method. However, among the methods that can split the computation, we obtain more accurate bounds as our bounds are more accurate per chunk.

**Reviewer 3:** *More experiments:* (a) As suggested by the reviewer, we used the training method in [2] to train an FFC NN with architecture 784-200-100-10 on MNIST for various values of $\lambda$ (a hyper-parameter controlling the Lipschitz constant of each layer). The figure on the right shows that `LipSDP` yields tighter bounds than those guaranteed by [2]. (b) We trained various NNs on MNIST with different initializations.

The variance of the Lipschitz bounds found by our method was essentially negligible. For instance, with one hidden layer and 200 neurons, the Lipschitz bounds were around 20 and the standard deviation over 6 trials was <0.01. *Naive lower bound:* In the special case of a purely linear network or under certain positivity assumptions, the "naive lower bound", defined in [4], is equal to the Lipschitz constant. By testing on randomly generated networks, it is our impression that this naive bound, introduced in [4], may be neither a lower bound nor an upper bound on the Lipschitz constant. We will clarify this in the revised manuscript. *Presentation:* We have improved the notation, wording, technical presentation, bibliography, and figures in the revised manuscript. For instance, (a) we clarified that $\Phi$ is "the

concatenation of activation functions at each layer"; (b) we made the presentation of the assumptions about the activation functions consistent and mentioned that [1] violates the assumption in our paper; (c) we updated the references and cited [1] and [2].

**References:** [1]: Anil, Cem, James Lucas, and Roger Grosse. "Sorting out Lipschitz function approximation." [2]: Gouk, Henry, et al. "Regularisation of neural networks by enforcing Lipschitz continuity." [3]: Dhillon, Guneet S., et al. "Stochastic activation pruning for robust adversarial defense."[4] Combettes, Patrick L., and J. Pesquet. "Lipschitz Certificates for Neural Network Structures Driven by Averaged Activation Operators." [5]: Lillicrap, Timothy P., et al. "Continuous control with deep reinforcement learning." [6]: Shi, Guanya, et al. "Neural lander: Stable drone landing control using learned dynamics.". [7]: Duan, Yan, et al. "Benchmarking deep reinforcement learning for continuous control."

[Meta-Review · NeurIPS 2019]

Congratulations on your work which the reviewers all appreciated. Please take the time to address their concerns for the final version of your paper.